

# Uncertainty analysis of hydrological return period estimation, taking the upper Yangtze River as an example

Hemin Sun[1,2], Tong Jiang[1,2], Cheng Jing[1,2], Buda Su[1,2,3], and Guojie Wang[1]

[1]Collaborative Innovation Center on Forecast and Evaluation of Meteorological Disasters, Nanjing University of Information Science & Technology, Nanjing, 210044, China
[2]National Climate Center, China Meteorological Administration, Beijing, 100081, China
[3]State Key Laboratory of Desert and Oasis Ecology, Xinjiang Institute of Ecology and Geography, Chinese Academy of Sciences, Urumqi, 830011, China

*Correspondence to*: Buda Su (subd@cma.gov.cn) and Guojie Wang (gwang_nuist@163.com)

**Abstract.** Return period estimation plays an important role in the engineering practices of water resources and disaster management, but uncertainties accompany the calculation process. Based on the daily discharge records at two gauging stations (Cuntan and Pingshan) on the upper Yangtze River, three sampling methods (SMs; (annual maximum, peak over threshold, and decadal peak over threshold), five distribution functions (DFs; gamma, Gumbel, lognormal, Pearson III, and general extreme value), and three parameterization methods (PMs; maximum likelihood, L-Moment, and method of moment) were applied to analyze the uncertainties in return period estimation. The estimated return levels based on the different approaches were found to differ considerably at each station. The range of discharge for a 20-year return period was 63,800.8–74,024.1 m3 s-1 for Cuntan and 23,097.8–25,595.3 m3 s-1 for Pingshan, when using the 45 combinations of SMs, DFs, and PMs. For a 1000-year event, the estimated discharge ranges increased to 74,492.5–125,658.0 and 27,339.2–41,718.1 m3 s-1 for Cuntan and Pingshan, respectively. Application of the analysis of variance method showed that the total sum of the squares of the estimated return levels increased with the widening of the return periods, suggestive of increased uncertainties. However, the contributions of the different sources to the uncertainties were different. For Cuntan, where the discharge changed significantly, the SM appeared to be the largest source of uncertainty. For Pingshan, where the discharge series remained almost stable, the DF contributed most to the uncertainty. Therefore, multiple uncertainty sources in estimating return periods should be considered to meet the demands of different planning purposes. The research results also suggest that uncertainties of return level estimation could be reduced if an optimized DF were used, or if the decadal peak over threshold SM were used, which is capable of representing temporal changes of hydrological series.

## 1 Introduction

A warming climate will cause increasingly frequent and more severe extreme events, exposing human society and ecosystems to greater climatic extremes (IPCC, 2014, Qin et al., 2015). Flooding and drought are among those extreme events that often cause huge damage. To cope with hydrological extremes, appropriate assessment of disaster and the



construction of hydraulic projects such as dams, bridges, and pipelines are greatly needed (Rosbjerg and Madsen, 1998; Milly et al., 2002; Cooley et al., 2007; Salvadori et al., 2011; Rootzén and Katz, 2013). Disaster assessment and mitigation relies heavily on the return period of hydrological extremes, where a T-year return period represents an event that has a 1/T chance of occurrence in any given year. However, because of limitations in understanding and data limitation, substantial

uncertainties accompany the fitting of extreme hydrological events, and it is necessary to quantify the uncertainties for robust decision making (Kay et al., 2006; Salvadori et al., 2011). Typically, an estimation of return period uses an appropriate parameterization method (PMs) to fit a candidate distribution for an extreme series. Uncertainty might stem from any procedure used in establishing the extreme series from the available data, selecting the distribution functions (DFs), or performing the parameterization process (Beck, 1987; Hoffman and Hammonds, 1994; El Adlouni, 2008; AghaKouchak and

Nasrollahi, 2010).

Since Horton (1896) used a normal distribution to fit hydrological extremes, many distribution families have been built. Their suitability as candidates can be evaluated by their ability to reproduce some important features of the data, e.g., upper/lower bounds of the distribution, left/right tails of the distribution, shape of the distribution, and exact zero values (Hosking, 1997). The frequency–magnitude distributions used most commonly in hydrological studies can be categorized

into four groups: the normal family (Normal, LogNormal(LN)), General Extreme Value (GEV) family (Gumbel (GUM), Fréchet, and Weibull), Pearson III (P III) family (Gamma (GAM), Pearson III, and Log-Pearson III), and Generalized Pareto distribution (Wakeby) (Cunnane, 1989; Beven and Binley, 1992; Hosking, 1997; El Adlouni et al., 2008; Heo et al., 2008; Malamud and Turcotte, 2006; Salinas et al., 2014). Two types of widely accepted sampling methods (SMs) are the annual maximum (AM) and peak over threshold (POT) methods. Studies have shown that the return levels estimated by POT series

are usually higher than by AM series (Fisher and Tippett, 1928; Pickands, 1975; Madsen et al., 1997a, b). Another key source of uncertainty that should not be neglected is related to the method of parameterization. Based on probability weighted moments, developed by Greenwood et al. (1979) for obtaining closed-form estimates of distribution parameters, Hosking (1990) found that certain linear combinations of probability weighted moments, named L-moments (LMs), could measure the location, scale, and shape of probability distributions. LMs form the basis of a comprehensive theory for the

description, identification, and estimation of distributions (Liu and Xu, 2015). Although moment-based methods are well established, it is sometimes difficult to assess the shape of a distribution conveyed by its moments, particularly when the sample is small (Hosking, 1997). Another common method is maximum likelihood (ML) estimation (Coles, 2001); however, it has a complex computation procedure.

Uncertainty assessment plays an important role in return-period-based decision making (Funtowicz and Ravetz, 1990). For

subjective method, such as the Pedigree Matrix (Barnett, 2006; Zhu et al., 2015), Carlo, Bayesian model averaging, and generalized likelihood methods (Vrugt et al., 2009; Tian et al., 2015), uncertainty estimation is often used for the approximation of the confidence intervals of quantiles of probability distributions (Coles and Pericchi, 2003; Parent and





Bernier, 2003; Beven, 2006; Blasone et al., 2008; Xu et al., 2010; Bouda et al., 2011; Delsman et al., 2013; Cheng et al., 2014a; 2014b; 2015). However, these methods actually only test the uncertainty from parameterization and they have the limitation of to be affected greatly by the prior probability. Recently, the analysis of variance (ANOVA) approach has been applied to the quantitative assessment of uncertainties in studies of climate change and climate change impact (Bosshard et al., 2013; Vetter et al., 2015). ANOVA is a model-based approach that partitions the total variance into components from different sources, allowing a fuller interpretation (Madden, 1976; Zwiers, 1987; 1996; Hingray et al., 2007). However, most current research focuses on the uncertainty analysis in one respect and few studies have considered the full scope of uncertainty during the process of return period estimation. Furthermore, although the hypothesis of stationarity in hydrological frequency analysis in a changing world has been questioned, it has not been given sufficient consideration in related studies (Milly, 2008; Salas and Obeysekera, 2013; Serinaldi and Kilsby, 2015).

The Cuntan and Pingshan hydrological stations on the Yangtze River (China) have experienced different levels of human interference. In this study, long-term records of observed discharge from these stations were selected for the estimation of the uncertainty sources in the process of deducing the return period based on combinations of five DFs (GAM, GUM, LN, GEV, and P III), three PMs (method of moment (MOM), ML, and LM), and three extreme SMs (AM, POT, and Decadal POT (DPOT)).

## 2 Data and methods

### 2.1 Data

The daily discharge records of the Cuntan and Pingshan hydrological stations on the Upper Yangtze River (Fig. 1), obtained from the "Hydrological Year Book-Yangtze" published by the Ministry of Water Resources in China, were selected for the analysis of the uncertainty in return period estimation (Jiang et al., 2007).

The drainage area of the Cuntan and Pingshan stations is about $8.6 \times 10^5$ and $4.8 \times 10^5$ km2, respectively. The discharge at Cuntan, averaged over the period 1939–2012, was $39.9 \times 10^5$ m3 s-1 and that at Pingshan, averaged over the period 1940–2011, was $16.5 \times 10^5$ m3 s-1. The maximum daily discharge was 84,300 (1 July 1981) and 28,600 m3 s-1 (16 September 1966) at Cuntan and Pingshan, respectively. Both stations have long-term observational records (over 70 years), but while the discharge at Cuntan has shown a significant downward trend, that at Pingshan has shown only slight changes (Fig. 2). Comparison of the records of the two stations can give us a clue that how does the different population series affect the return period estimation results.



## 2.2 Sampling Methods

Three SMs were considered in this study. One is block maximum approach, which consists of selected maximum within a block. Here, a one-year block was used to choose the annual maxima (AM) series. The second was the POT method, which chooses all extremes that exceed a given threshold (Davison and Smith, 1990; Madsen et al. 1997a; 1997b). Furthermore,

considering the possibility that an extreme series might be changed significantly in a changing world, we established and applied the DPOT approach, which considers a 10-year period (i.e., a decade) as a block and applies the POT method to each block to sample the extremes.

## 2.3 Distribution Functions

Skewness and kurtosis are indicators of the shape of a sample series distribution. Skewness is the third moment of a data

series and it represents by how far the series deviates from a normal distribution. If skewness = 0, the empirical probability density function (EPDF) follows a normal distribution; if skewness > 0, the right-hand tail of the EPDF will be longer and become fatter as skewness increases (Groeneveld and Meeden, 1984; von Hippel, 2005; Cao et al., 2013). Kurtosis is the fourth moment of a data series and it represents the peakedness of the distribution. For a normal distribution, kurtosis = 3; a distribution with a lower value of kurtosis has a heavy tail, which denotes that greater variance is attributable to frequent

extreme deviations (Westfall, 2014).

Considering different climatic and environmental backgrounds, regional optimistic distributions are used in different countries to analyze the frequency of hydrological extremes. For example, the Log-Pearson III distribution is recommended by the United States and Australia, the GUM distribution has been used in Canada and India, and the LN and Wakeby distributions have been applied in Japan and Korea (Water Resources Council, USA, 1982; Park et al., 2001; Öztekin, 2007;

El Adlouni et al., 2008;). Since the 1960s, the P III distribution, which is recommended by the Chinese Ministry of Water Resources (1995), has been used commonly in flood frequency analysis in China. Therefore, in this study, five distributions used worldwide in hydrological frequency analysis, namely GEV, P III, GAM, LN, and GUM were chosen to fit the extreme discharge series of the upper Yangtze River. GAM, LN, and GUM are two-parameter distributions, whereas GEV and P III are three-parameter distributions. Their probability density functions (PDFs) f(x) and their cumulative distribution functions

(CDFs) F(x) are shown in Table 1.

## 2.4 Parameterizations

Different PMs might result in diverse quintile estimations. To evaluate the range of return levels using different parameterizations, three conventional approaches were applied to estimate the unknown parameters. These three methods are described briefly in the following.





### 2.4.1 MOM

The shape of a probability distribution has traditionally been described by the moments of the distribution. Analogous quantities can be computed from an independent and identically distributed data sample $\{x_1, x_2,\ldots, x_n\}$. The moments are

$$m_r^{'} = \frac{1}{n}\sum_{i=1}^{n}x_i, m_1^{'} = \bar{x} = mean , \tag{1}$$

and

$$m_r = \frac{1}{n}\sum_{i=1}^{n}\left(x_i - \bar{x}\right)^r, m_1 = 0 , \tag{2}$$

where $n$ is the sample size.

The coefficient of variation ($C_V$) is

$$C_v = z = \frac{m_2^{1/2}}{m_1^{'}} \tag{3}$$

The coefficient of skewness ($C_S$) is

$$C_s = \gamma_1 = \frac{n^2}{(n-1)(n-2)}\frac{m_3}{m_2^{3/2}} . \tag{4}$$

### 2.4.2 ML

It is desirable to find an estimator $\theta$ as close to the true value $\theta_0$ as possible. Thus, $\theta$ is the function's variable and it is allowed to vary freely. The function of the likelihood can be structured as

$$L\left(x_1, x_2,\ldots, x_n;\theta\right) = \prod_{i=1}^{n} f\left(x_i;\theta\right) \quad \left(\theta \subseteq \varnothing\right). \tag{5}$$

In practice it is often more convenient to work with the logarithm of the likelihood function, called the log-likelihood:

$$\ln L\left(x_1, x_2,\ldots, x_n;\theta\right) = \prod_{i=1}^{n} \ln f\left(x_i;\theta\right). \tag{6}$$

### 2.4.3 LM

In terms of probability weighted moments, LMs are given by:

$$\begin{cases} l_1 = b_0 \\ l_2 = 2b_1 - b_0 \\ l_3 = 6b_2 - 6b_1 + b_0 \\ l_4 = 20b_3 - 30b_2 + 12b_1 - b_0 \end{cases}, \tag{7}$$



where $b_0 = \dfrac{1}{n}\sum_{i=1}^{n} x_i$ , $b_1 = \dfrac{1}{n}\sum_{i=2}^{n} \dfrac{i-1}{n-1} x_i$ , $b_2 = \dfrac{1}{n}\sum_{i=3}^{n} \dfrac{(i-1)(i-2)}{(n-1)(n-2)} x_i$ , and $b_3 = \dfrac{1}{n}\sum_{i=4}^{n} \dfrac{(i-1)(i-2)(i-3)}{(n-1)(n-2)(n-3)} x_i$ .

The LM ratios are defined as follows:

$$t_2 = l_2 / l_1 , \quad t_3 = l_3/l_2 , \quad \ldots , \quad t_r = l_r / l_2 \quad (r = 3,4,\ldots) , \tag{8}$$

where $t_2$ is $L\text{-}C_v$, $t_3$ is L- skewness, and $t_4$ is $L$- Kurtosis (Hosking 1990; 1997).

**2.5 Goodness of Fit Test**

The Chi-squared ($\chi^2$) statistic is used to examine the distances between the distributions of features, and figures out if a sample comes from a hypothesized continuous distribution (Stanberry, 2013; Corder and Foreman, 2014). The hypothesis regarding the distributional form is rejected at the chosen significance level ($\alpha$) if the test statistic is greater than the critical value. For a random sample $\{x_1, \ldots , x_n\}$, the following formula is used to define the number of bins (k):

$$k = 1 + \log_2 N . \tag{9}$$

Then, the Chi-squared statistic can be defined as

$$\chi^2 = \sum_{i=1}^{k} \frac{(O_i - E_i)^2}{E_i} , \tag{10}$$

where $O_i$ is the observed frequency for bin $i$, and $E_i$ is the expected frequency for bin $i$, which can be calculated by

$$E_i = F(x_2) - F(x_1) , \tag{11}$$

where $F$ is the CDF of the probability distribution being tested, and $x_1$ and $x_2$ are the limits for bin $i$.

**2.6 Method of Uncertainty Assessment**

The ANOVA method was used to partition the variance of uncertainty (Von Storch and Zwiers, 2001; Deque et al., 2007; Yip et al., 2011). The framework of the three-way ANOVA used in the current study is depicted in Fig. 3. The different sample sizes of the uncertainty sources (three SMs, five DFs, and three PMs) might result in a biased variance estimation,

which can be avoided using a subsampling scheme (Bosshard et al., 2013; Vetter et al., 2015). Here, subsampling was conducted to guarantee that each subsample had three distributions, three sample methods, and three parameter methods (3 × 3 × 3). The unbiased variance fractions $\eta^2$ related to different components can be calculated as

$$\eta_x^2 = \frac{1}{10}\sum_{d=1}^{10} \frac{SS_x(m)}{TSS(m)} , \tag{12}$$

where $TSS(m)$ is the total sum of the squares ($TSS$) of each subsample. The $TSS$ is defined as

$$TSS = \sum_{i=1}^{3}\sum_{j=1}^{3}\sum_{k=1}^{5} \left( Y_{ijk} - \overline{Y}_{ooo} \right)^2 , \tag{13}$$





where $Y_{ijk}$ is the particular value corresponding to SM $i$, PM $j$, and DF $k$, respectively; and $Y_{ooo}$ is the overall mean. The ANOVA can split the total sum of the squares into the sums of the squares due to the individual effects ($SS_{Data}$, $SS_{Param}$, $SS_{Dis}$).

## 3 Results

### 3.1 Statistical Characteristics of Extreme Series

The EPDFs of AM, POT, and DPOT in the upper Yangtze River are shown in Fig. 4. For Cuntan, the skewness and kurtosis of the AM, POT, and DPOT series are 0.40 and 0.41, 1.76 and 5.25, and 1.12 and 3.10, respectively. The EPDF of the AM series has the heaviest tail and that of the POT series has the longest tail.

Discharge at Pingshan has no significant trend over time. The kurtosis of its extreme series does not show any significant difference among the three SMs. The Skewness of its AM series is 0.61 (slightly higher than Cuntan) and its kurtosis is 0.48 (comparable with Cuntan). The skewness and kurtosis of Pingshan's POT series are 1.21 and 1.85, respectively; much lower than Cuntan. The skewness and kurtosis of the DPOT series are 0.68 and 0.81, respectively; the kurtosis is lower than Cuntan.

### 3.2 Goodness of Fit of Distributions

The χ2 test was applied to examine the goodness of fit (GOF) of the five distributions for fitting the upper Yangtze River hydrological extremes. A low value of the χ2 statistic indicates smaller differences between the theoretical PDF and EPDF. The results of the χ2 test statistics for the 45 combinations of SMs, DFs, and PMs are shown in Fig. 5. For the Cuntan discharge, except for the fitting of GEV to the POT series by the ML method, which fails to pass the χ2 GOF test (at the 0.05 significance level), all the fitting results from the other 44 combinations are satisfactory (Fig. 5a). For the Pingshan discharge, all 45 results pass the χ2 test at the 0.05 significance level (Fig. 5b).

The variances of the fitting results of the SMs are shown in the averaged χ2 statistics of the 15 combinations fitted by the different DFs (GAM/GUM/LN/P III/GEV) and PMs (ML/LM/MOM) for each SM in Fig. 5a and 5b. It is clear that for both stations the statistics of the DPOT-based combinations are smallest. This suggests that the DPOT sequence can be described well by all five selected DFs using all three PMs, but that the optimum way to deduce return levels for the DPOT series for both stations is the LM–GAM combination. For the AM series, the combinations of LM–GAM and ML–P III are optimum for Cuntan and Pingshan, respectively. For the POT series, the ML–P III combination is optimum for both stations. For all three extreme series at Pingshan, the fitting results from the combination of the LN distribution using LM parameterization are the worst. For the Cuntan extremes, the LM–LN combination is the worst choice for the AM series, ML–GEV is the worst for the POT series, and MOM–GAM is the worst for the DPOT series.



To illustrate the effects of the DFs on the fitting results, the averaged χ2 statistics of nine combinations of SMs (AM/POT/DPOT) and PMs (ML/LM/MOM) fitted by each function are shown in Fig. 5c and 5d. Of the five distributions, the three-parameter distributions (P III and GEV) fit better than the two- parameter distributions (LN, GUM, and GAM) and the fitting result of the LN distribution are the worst. For the Cuntan extreme series, GAM is the optimum distribution, followed by the P III distribution. The averaged χ2 statistics of the GEV-based combinations are higher than the GAM-based combinations, and except for the poorer fit of the GEV–POT–ML combination, the GEV-based fittings are even better than the GAM-based results. For the Pingshan extreme series, P III is the optimum distribution, followed by the GEV distribution. The LN and GUM distributions do not fit the POT series well.

The variabilities of the averaged χ2 statistics for the three PMs are relatively smaller than for the SMs and DFs. The χ2 statistics of the LM-based fitting results are higher than the ML- and MOM-based fittings, and this variance is higher at Pingshan than Cuntan (Fig. 5c and 5d). For the Cuntan extreme series, the χ2 statistics are smallest for GAM–DPOT, P III–POT, and LN–DPOT using LM, ML, and MOM, respectively. The largest χ2 statistics are obtained using the LN–AM, GEV–POT, and GUM–AM combinations with the LM, ML, and MOM methods, respectively. For the Pingshan extreme series, the smallest χ2 statistics are derived with P III–POT using the ML and LM methods, and with the GAM–DPOT combinations using MOM. The largest χ2 statistics are obtained with LN–POT combinations using ML and LM and by GAM–POT combinations using MOM (Fig. 5a and 5b).

### 3.3 Uncertainty Range of Return Period Events

The estimation of T-year return period events using all 45 combinations of the 3 SMs, 5 DFs, and 3 PMs shows that large variability can be found in the return levels. Figure 6 indicates that the range of return levels by the nine LN-based combinations is the largest, while that of the nine P III-based combinations is the smallest among the five distributions used. Note that because of the poor performance of the POT–ML–GEV combination, the range of return levels estimated by the GEV distribution is also very large for the Cuntan extremes. The return levels estimated by the DPOT method show steady performance for all PM and DF combinations, with the smallest range of return levels compared with other SMs. The return levels estimated by the three PMs are very similar.

For Cuntan, the 20-year return period event deduced by the 45 combinations ranges from 63,800.8 to 74,024.1 m$^3$ s$^{-1}$; the variance of 10,223.3 m$^3$ s$^{-1}$ is about 15% of the average level (66,808.2 m$^3$ s$^{-1}$). The 50-year event ranges from 66,716.2 to 83,115.0 m$^3$ s$^{-1}$; the variance of 16,398.8 m$^3$ s$^{-1}$ is 23% of the average level (71,995.1 m$^3$ s$^{-1}$). The 100-year event ranges from 68,707.4 to 89,787.8 m$^3$ s$^{-1}$; the variance of 21,080.4 m$^3$ s$^{-1}$ is 28% of the average level (75,845.2 m$^3$ s$^{-1}$). The 200-year event ranges from 70,563.4 to 97,661.5 m$^3$ s$^{-1}$; the variance of 27,098.1 m$^3$ s$^{-1}$ is about 34% of the average level (79,617.8 m$^3$ s$^{-1}$). The 500-year event ranges from 72,855.9 to 112,444.0 m$^3$ s$^{-1}$; the variance of 39,588.1 m$^3$ s$^{-1}$ is 47% of the average level



(84,278.7 m$^3$ s$^{-1}$). The 1000-year event ranges from 74,492.5 to 125,658.0 m$^3$ s$^{-1}$; the variance of 51,165.5 m$^3$ s$^{-1}$ is 58% of the average level (88,240.2 m$^3$ s$^{-1}$) (Fig. 6a).

The estimated ranges of return level events at Pingshan are smaller than at Cuntan. The 20-year return period event estimated by the 45 combinations ranges from 23,097.8 to 25,595.3 m$^3$ s$^{-1}$; the variance of 2497.5 m$^3$ s$^{-1}$ is about 11% of the average return level (23,761.3 m$^3$ s$^{-1}$). The 50-year event ranges from 24,250.6 to 28,740 m$^3$ s$^{-1}$; the variance of 4489.4 m$^3$ s$^{-1}$ is about 17% of the average level (25,710.4 m$^3$ s$^{-1}$). The 100-year event ranges from 25,039.6 to 31,048.3 m$^3$ s$^{-1}$; the variance of 6008.7 m$^3$ s$^{-1}$ is about 22% of the average level (27,123.7 m$^3$ s$^{-1}$). The 200-year event ranges from 25,776.2 to 33,323.0 m$^3$ s$^{-1}$; the variance of 7546.8 m$^3$ s$^{-1}$ is about 26% of the average return level (28,502.0 m$^3$ s$^{-1}$). The 500-year event ranges from 26,687.6 to 37,604.3 m$^3$ s$^{-1}$; the variance of 10,916.7 m$^3$ s$^{-1}$ is about 36% of the average level (30,288.1 m$^3$ s$^{-1}$). The 1000-year event ranges from 27,339.2 to 41,718.1 m$^3$ s$^{-1}$; the variance of 14,378.9 m$^3$ s$^{-1}$ is about 45% of the average level (31,621.9 m$^3$ s$^{-1}$) (Fig. 6b).

### 3.3.1 Return Level Variation based on Different Samplings

To analyze the range of different SMs, return period events estimated by the combinations of the five DFs (GEV/P III/GAM/GUM/LN) and three PMs (ML/LM/MOM) were averaged for each SM (Fig. 7a1 and 7b1). The range of averaged return levels estimated using the different SMs shows an increasing trend from shorter to longer return period events.

For the Cuntan discharge, the average return levels of 20–1000-year events are 68,085.71–92,436.35, 65,862.59–87,633.27, and 66,176.39–~84,654.03 m$^3$ s$^{-1}$ for the AM, POT, and DPOT series, respectively. The range of return levels estimated by the different SMs is 2223.1–7785.3 m$^3$ s$^{-1}$ for the 20–1000-year events, which is about 3%–8% of the averaged return level over the three SMs (Fig. 7a1). For the Pingshan discharge, the average return levels of 20–1000-year events are 23,675.05–32,755.79, 23,688.6–31,304.66, and 23,920.25–30,805.37 m$^3$ s$^{-1}$ based on AM, POT, DPOT samples, respectively. The changes of estimated return period extremes by the different SMs at Pingshan are obviously smaller than at Cuntan, with values ranging from 403.7 to 1324.6 m$^3$ s$^{-1}$ for 20–1000-year events, which are about 2%–4% of the averaged return level over the three SMs (Fig. 7b1).

### 3.3.2 Return Level Variation based on Different Parameterizations

Return period events estimated by the combinations of five DFs (GAM/GUM/LN/P III/GEV) and three SMs (AM/POT/DPOT) were averaged over each PM (Fig. 7a2 and 7b2).

For the Cuntan discharge, the return levels by the three PMs are similar, ranging between 697.8 and 2360.4 m$^3$ s$^{-1}$ for 20–1000-year events, which are about 1% and 3% of the averaged return levels over the three PMs. The average of return period events of MOM-based combinations is the smallest among the three parameterizations, with a value of 66,359.6–86,978.6 m$^3$ s$^{-1}$ for 20–1000-year events. The return levels estimated by LM are the largest for events shorter than 200 years, whereas



those estimated by ML-combinations are the largest for events longer than 200 years. The average of the 15 LM- and ML-based combinations is 67,057.4–88,403.1 and 66,707.7–89,399.0 m$^3$ s$^{-1}$ for 20- and 1000-year events, respectively (Fig. 7a2).

For the Pingshan series, the ranges of the return levels derived by the three PMs are larger than for Cuntan, with values of 245.2–1950.4 m$^3$ s$^{-1}$ for 20–1000-year events, respectively, i.e., about 1%−6% of the averaged return levels over the three PMs. The average return levels estimated by MOM are also the smallest, with values of 23,920.3–30,805.4 m$^3$ s$^{-1}$ for 20–1000-year events, while the LM-based combinations have the largest average return levels with values of 23,675.1–32,775.8 m$^3$ s$^{-1}$ for 20–1000-year events. For the ML-based combinations, the average return levels of all 15 LM-based combinations are 23,688.6–31,304.66 m$^3$ s$^{-1}$ for 20–1000-year events (Fig. 7b2).

### 3.3.3    Return Level Variation based on Different Distributions

The range of averaged return levels of the nine combinations of three SMs (AM/POT/DPOT) and three PMs (ML/LM/MOM) over each of the five DFs is greater than for the PM- and SM-based results (Fig. 7a3 and 7b3).

For the Cuntan discharge, the results of GUM-based combinations have the largest return levels among the five functions, with values of 66,952.6–92,751.8 m$^3$ s$^{-1}$ for 20–1000-year events. The average return levels of the GAM-based combinations are the smallest, with values ranging from 65,532.9–80,719.4 m$^3$ s$^{-1}$ for 20–1000-year events. The range of the five DFs is between 1767.4 and 12,032.4 m$^3$ s$^{-1}$ for 20–1000-year events, which is about 3%−14% of the averaged return level over the five DFs (Fig. 7a3).

A similar pattern is found for the Pingshan extremes. The results of the GAM-based combinations have the smallest return levels for all five functions, with values of 23,344.6–29,009.2 m$^3$ s$^{-1}$ for 20–1000-year events. The return levels of the GUM-based combinations have the largest average return levels, with values of 23,860.6–33,415.8 m$^3$ s$^{-1}$ for 20–1000-year events. The range of the five DFs is between 664.7 and 4406.6 m$^3$ s$^{-1}$ for 20–1000-year events, which is about 3%−14% of the averaged return level over the five DFs (Fig. 7b3).

### 3.4 Evaluation of Uncertainty Contribution

The ANOVA method was used to decompose the total uncertainty into different sources. The total sum of the squares increases with increasing return period, but the contributions from the different sources to the overall uncertainty might change over the return period. Figure 8 shows the contributions from the different sources (SMs, DFs, and PMs) to the overall uncertainty.

For the Cuntan discharge, the SMs contribute more than 40% of the total uncertainty in the estimation of return levels, and this changes little for 20–1000-year event estimations. The DFs also exert obvious influence on the return period estimation,





contributing about 30%–40% of the total uncertainty, and this uncertainty increases with time. The PMs contribute the least to the uncertainty, accounting for about 10%–20% of the total.

A different pattern can be seen for the uncertainty of return period estimation at Pingshan. The DFs are the dominant source, explaining almost 50% of the total uncertainty, and this is constant over time. The SMs contribute about 22%–35% of the total uncertainty, and this proportion increases with the return period. Uncertainty originating from the PMs is the least important individual source of uncertainty at Pingshan (but more pronounced than that at Cuntan), contributing about 17%–33% of the total uncertainty, and this proportion decreases with the return period.

## 4 Discussion and Concluding Remarks

This study used daily discharge monitored at the Cuntan (which had a significant trend of discharge) and Pingshan (which had no obvious trend of discharge) hydrological stations on the upper Yangtze River in China to compare the return period events estimated by different SMs (AM/POT/DPOT), DFs (GAM/GUM/LN/GEV/P III) and PMs (ML/LM/MOM). The contributions of the SMs, PMs, and DFs to the total uncertainty of the return period estimations was analyzed using the ANOVA method.

A large variability of return period events was estimated by the 45 different combinations. For Cuntan, the range of return levels was about 15% of the average (66,808.2 $m^3$ $s^{-1}$) for 20-year events and up to 58% of the average (88,240.2 $m^3$ $s^{-1}$) for 1000-year events. The ranges of return levels estimated by all 45 combinations at Pingshan were smaller than at Cuntan. For 20–1000-year events, the ranges of return levels were about 11%–45% of the average return level (23,761.3–31,621.9 $m^3$ $s^{-1}$).

The selection of SMs, DFs, and PMs clearly affected the return period estimations. The ranges of the averaged return levels of the three different PMs were similar, but those of the five DFs were the largest. The SMs caused intermediate variation of the return levels but with larger differences at Cuntan than at Pingshan.

The ranges of return levels estimated by two-parameter DFs (GAM, GUM, and LN) were larger than estimated by three-parameter DFs. According to the GOF test, the worst fitting LN distribution had the largest range of return levels, whereas the best fitting P III distribution had the smallest range. Note that because of the poor performance of the POT–ML–GEV combination, the range of return levels estimated with the GEV distribution was also very large for the Cuntan extremes. This indicates that it is not only the individual factors but also the interaction of the SMs, PMs, and DFs that affects the uncertainty of the return period estimation. The return levels estimated by the DPOT method demonstrated steady performance for every combination of PM and DF, with the smallest range of return levels in comparison with the other SMs.

The results of the ANOVA analysis showed that all modeling chain combinations had an increasing trend of the total sum of the squares with increasing return periods, but that the contributions of the different sources were different. Although the



largest range of return levels stemmed from the DFs for both the Cuntan and the Pingshan discharge series, as the number of DFs used in this study was greater than the SMs and PMs, after the subsampling scheme, the main sources of uncertainty associated with return level estimation at the different stations were different. For Cuntan, which showed obvious change, the SMs had the greatest influence on the return levels, contributing more than 40% of the total uncertainty; however, for

Pingshan, the DF was the most important source of uncertainty, contributing almost 50% of the total. The proportion of the contribution of the PMs to the total uncertainty is less significant than that of the SMs and DFs, but it plays a more important role in the Cuntan discharge series than the Pingshan series. This indicates that the sources of uncertainty in the return period estimations are different for discharges with and without significant trends, and that multiple uncertainty sources in the estimations of return periods should be considered to meet the demands of different planning purposes.

Through a case-dependent uncertainty analysis in this study, it was found that the selection of different methods has significant effects on return period estimations. An estimation return period is an approximated figure and thus, some degree of uncertainty remains in the process. There are two possible ways to reduce the associated uncertainty. The first is to obtain a sufficiently long and high-quality extreme series, as in Parent and Bernier (2003), who highlighted that the incorporation of historical records and regional information into the process of deducing return periods is useful for uncertainty reduction.

Unfortunately, reliable data are not always available, and therefore choosing better SMs is the more viable alternative. This study found that of the three SMs the DPOT method brought the least variation. The ranges of return levels based on the DPOT series were usually the smallest compared with the other two sampling series, and therefore it was considered suitable for sampling discharge time series with or without significant trends. Furthermore, it also performed best when applying different DFs and PMs. Several studies have recommended a nonstationary method to reduce the uncertainties in a changing

world, by applying time-dependent distribution parameters. However, the application of these models remains immature and the hypothesis that "stationarity is dead" remains debatable (Milly et al., 2008; Salas and Obeysekera, 2013; Cheng et al., 2014; Serinaldi and Kilsby, 2015). For a station that has significant trends, the application of DPOT sampling might be one way to reduce uncertainty in return period estimations.

The second option is to reduce the uncertainty in the return period estimation processes is using more candidate DFs, and

then by choosing an optimum DF to reduce the total uncertainty. In this study, the P III distribution performed well at both stations and the ranges of the return levels of the P III-based combinations were the smallest. However, it must be noted that the DF itself, as well as the interaction of the DF with the SMs and PMs, contributes to the uncertainty in the return period estimation. The GEV is recommended by the WMO to fit annual climate extremes (Klein Tank et al., 2009). However, it failed to pass the GOF test when fitting the Cuntan POT series by the MLE method, although other GEV-based

combinations did fit the extreme discharge series very well (Botero and Francés, 2010; Salinas et al., 2014). In addition, Hosking (1997) indicated that the return period estimated by two-parameter distributions would be accurate only when the fitted distribution approximated the true distribution; otherwise, it could be severely biased. The results of this study also



suggested that the ranges of return periods estimated by the three-parameter distributions were smaller than those of the two-parameter distributions, and that the three-parameter distributions were more reliable in the frequency analysis of extremes. Therefore, results derived from two-parameter DFs should be treated with caution in practice.

As mentioned in the introduction, many distributions have been used in return period estimations (Park et al., 2001; Öztekin, 2007; Su et al., 2009; Fischer et al., 2012). The selection of only five distributions is one of the primary limitations of the current study. The uncertainty ranges enlarged with increasing return periods. To achieve different goals, different sources of uncertainty should be considered when estimating return levels, especially with respect to the desired frequency when estimating the return levels. Considering the weights of uncertainties in return level estimations, reasonable SMs and DFs should be selected carefully in future hydraulic design and risk management practices, according to the level of return periods required.

**Acknowledgments**

This research was supported by the National 1000 Talent program (Y474171), the National Basic Research Program of China (973 Program) (2013CB430205), the National Natural Science Foundation of China (41571494, 41375099, 91337108, 41561124014). Many thanks to the anonymous reviewers who supplied useful suggestions and helped to significantly improve this manuscript. Furthermore, we are thankful to J. L. Huang for valuable discussions, and T. Vetter for introduction of the ANOVA method.

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




**Table 1**. Distribution functions fitted to hydrological samples from the Cuntan and Pingshan hydrological stations

| Distribution Name | Probability Density Function | Cumulative Distribution Function |
|---|---|---|
| **Gamma (Gam)** $\alpha, \beta (\alpha > 0, \beta > 0)$ | $f(x) = \dfrac{x^{\alpha-1}}{\beta^{\alpha}\Gamma(\alpha)} \exp\left(-\dfrac{x}{\beta}\right)$ | $F(x) = \dfrac{\Gamma_{x/\beta}(\alpha)}{\Gamma(\alpha)}$ |
| **Gumbel (Gum)** $\sigma, \mu (\sigma > 0)$ | $f(x) = \dfrac{1}{\sigma} \exp(-z - \exp(-z))$ | $F(x) = \exp(-\exp(-z))$ |
| **Lognormal (LN)** $\sigma, \mu (\sigma > 0)$ | $f(x) = \dfrac{\exp\left(-\dfrac{1}{2}\left(\dfrac{\ln x - \mu}{\sigma}\right)^2\right)}{x\sigma\sqrt{2\pi}}$ | $F(x) = \Phi\left(\dfrac{\ln x - \mu}{\sigma}\right)$ |
| **Pearson-Ⅲ (P Ⅲ)** $\alpha, \beta, \gamma (\alpha > 0, \beta > 0)$ | $f(x) = \dfrac{(x-\gamma)^{\alpha-1}}{\beta^{\alpha}\Gamma(\alpha)} \exp\left(-\dfrac{x-\gamma}{\beta}\right)$ | $F(x) = \dfrac{\Gamma_{(x-\gamma)/\beta}(\alpha)}{\Gamma(\alpha)}$ |
| **Generalized Extreme Value (GEV)** $k, \sigma, \mu (\sigma > 0)$ | $f(x) = \begin{cases} \dfrac{1}{\sigma} \exp\left(-(1+kz)^{-1/k}\right)(1+kz)^{-1-1/k} & k \neq 0 \\ \dfrac{1}{\sigma} \exp(-z - \exp(-z)) & k = 0 \end{cases}$ | $F(x) = \begin{cases} \exp\left(-(1+kz)^{-1/k}\right) & k \neq 0 \\ \exp(-\exp(-z)) & k = 0 \end{cases}$ |

Note: $z \equiv \dfrac{x-\mu}{\sigma}$; $\Gamma(\alpha) = \int_{0}^{\infty} t^{\alpha-1} e^{t} dt$ $(\alpha > 0)$; $\Gamma_{x}(\alpha) = \int_{0}^{x} t^{\alpha-1} e^{-t} dt$ $(\alpha > 0)$




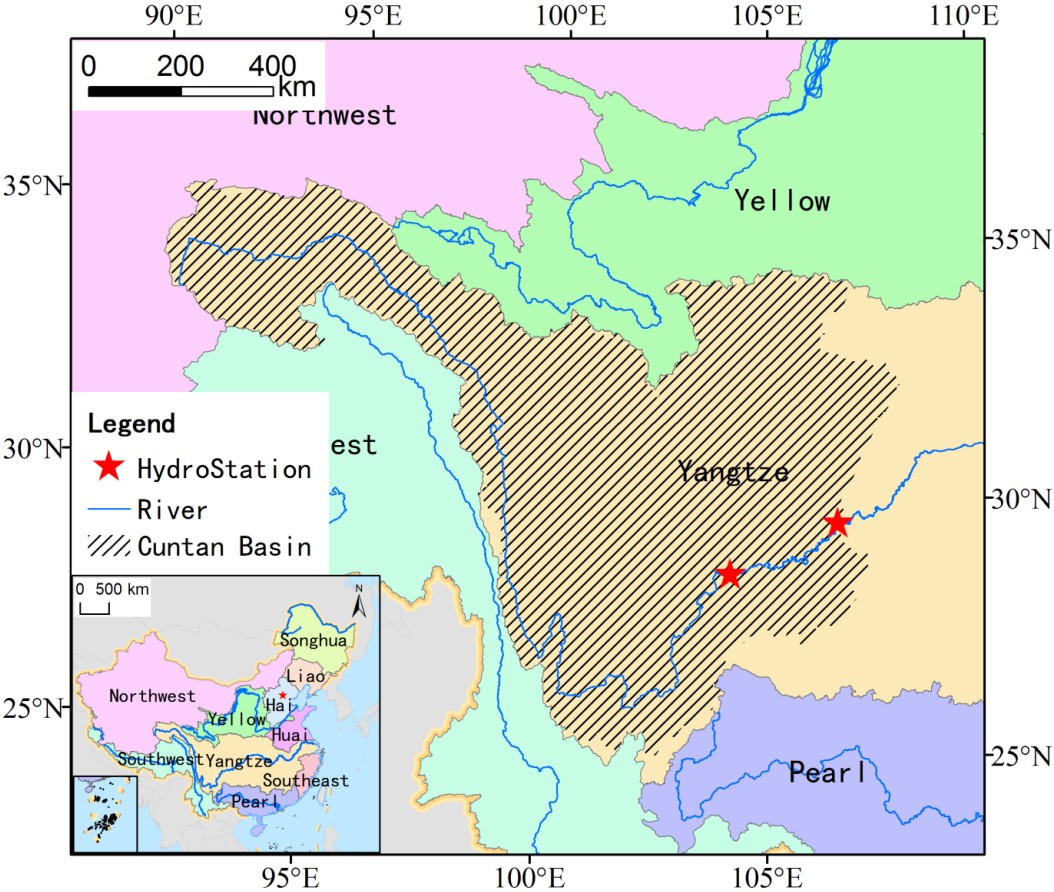

**Figure 1:** Locations of the hydrological stations in the Pingshan and Cuntan River basins on the upper Yangtze River, China.





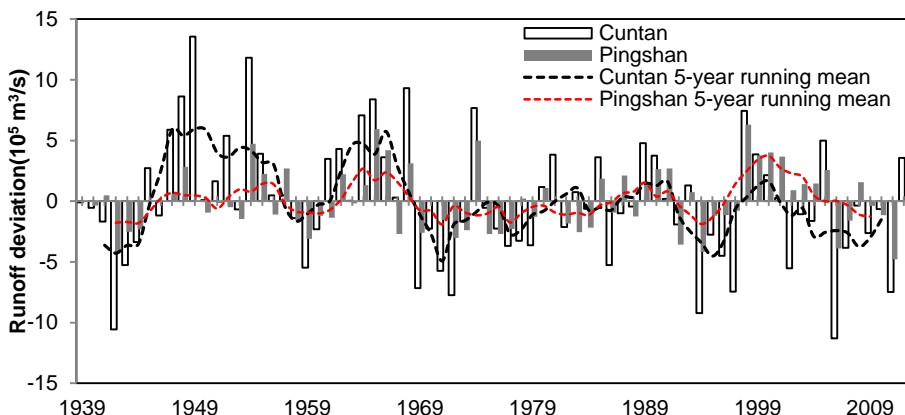

**Figure 2:** Annual total runoff at the Pingshan (gray histogram) and Cuntan (black histogram) hydrological stations. The black line represents the 5-year running mean at Cuntan. The red line represents the 5-year running mean at Pingshan.





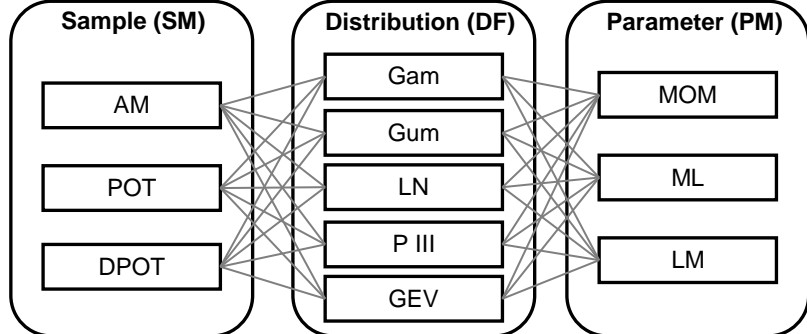

**Figure 3:** Modeling chain combination scheme. The three analyzed modeling chain elements are depicted from left to right.




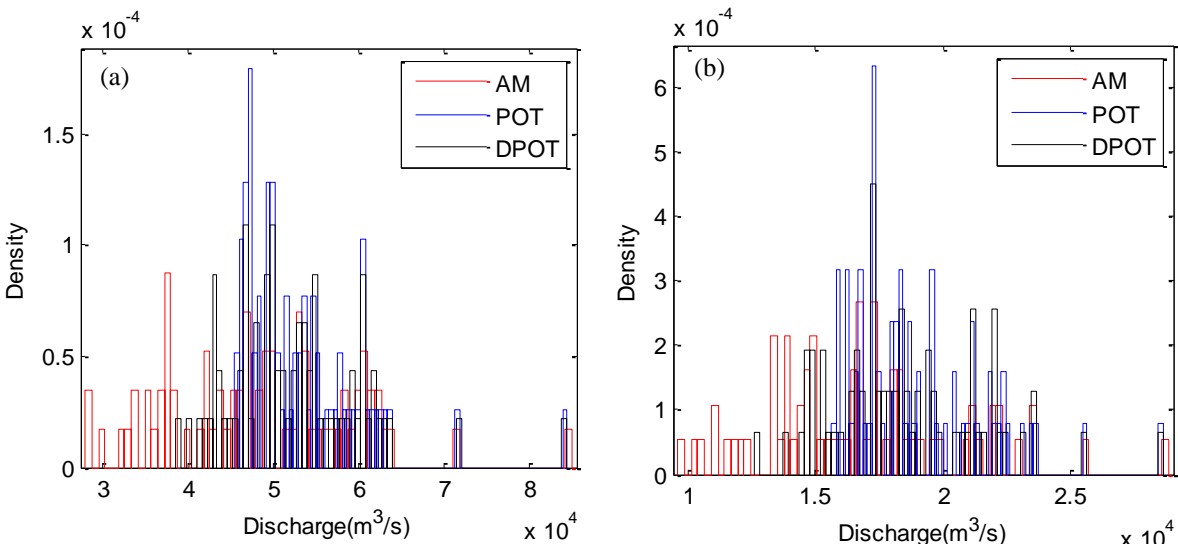

**Figure 4:** Three sample method frequency histogram for the extreme discharge series: (a) Cuntan and (b) Pingshan. Red,

blue, and black histograms represent the AM, POT, and DPOT extreme discharge series, respectively.





**Figure 5:** $\chi^2$ goodness of fit tests of different distributions: (a) and (c) Cuntan, and (b) and (d) Pingshan. Black line indicates the

5    0.05 significance level of $\chi^2$ goodness of fit tests.





**Figure 6:** CDF of all combinations: (a) Cuntan and (b) Pingshan. 1–5 represent the CDFs of the GAM, GUM, LN, GEV, and P III DFs, respectively.





**Figure 7:** Average return levels to extreme discharge: (a) Cuntan and (b) Pingshan. 1−3 represent the average return levels

5    estimated by different SMs, PMs, and DFs.





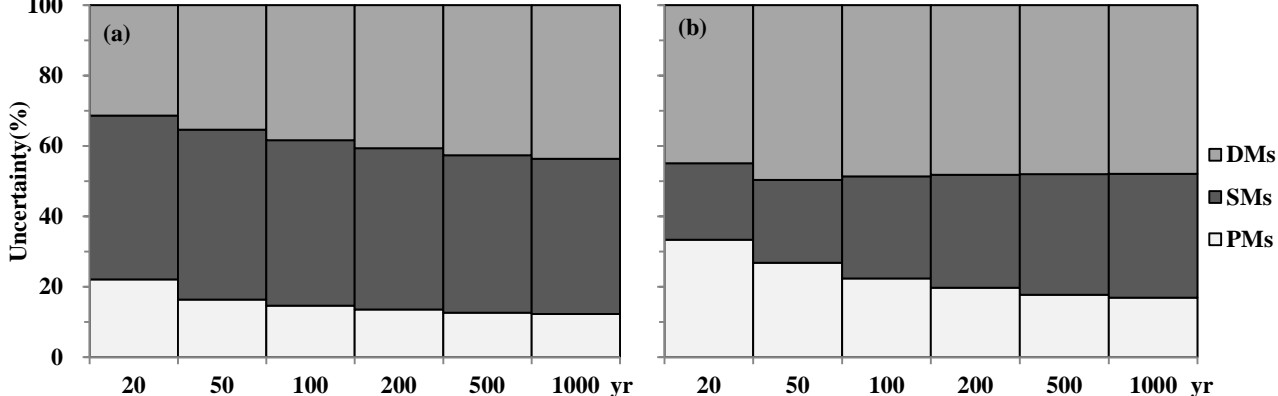

**Figure 8:** Contributions of different sources of uncertainty to overall uncertainty with subsampling scheme: (a) Cuntan and

(b) Pingshan.