# Peer review of "Uncertainty analysis of hydrological return period estimation, taking the upper Yangtze River as an example"

_Hydrology and Earth System Sciences, 2016_

## Referee Comment (RC1) · Anonymous Referee #1 · 16 Mar 2017

This paper addresses the uncertainty in estimating return periods by considering different data sampling approaches, distribution assumptions and parameter estimation methods. It showed that different approaches could lead to very different results, and the optimal approach varies across gauging stations. The work is technically sound and the manuscript is clearly organized. The results are of practical importance.

Major comments: 1) The study somewhat lacks an in-depth discussion. The results are case-dependent and do not have a general implication. One reason is that it considered only two gauging stations, and they are in the similar climate and watershed conditions. They do differ in the flow variation trend, the uncertainty results reflect the difference. However, the author didn't go further to reveal de underlying physical or

mathematical reasons for the difference. Thus, no general conclusion can be drawn from the comparison. I improve the scientific significance of this work, I suggest the authors either analyze more stations in different watershed and climate conditions, or provide a theoretical analysis of the difference between the two stations. 2) The introduction to the sampling methods in Section 2.2 is two succinct. With the limited information, readers may not be able to understand how the extreme series are actually produced through POT and DPOT. Missing such critical information makes it hard for readers to understand the work.

Minor comments: 1) Acronyms are not consistent in the text and figures. Some examples are: MLE vs. ML; P III vs. P3; LN vs. LN2; Gam vs. GAM; GUM vs. GUM. . . 2) Section 3.3 was poorly written. It pours a lot of numbers here, but provides few insights. This section could be condensed into a couple of tables or figures, following by a paragraph os summary. 3) Please check the units, many of them do not have correct superscripts. 4) The introduction should articulate the research objectives.

---

## Referee Comment (RC2) · Anonymous Referee #2 · 29 Mar 2017

This paper introduces and describes a quantitative way to assess the uncertainty in estimating return periods by considering different data sampling methods, distribution functions and parameterizations estimation methods. The topic of this manuscript had practical value for the engineering design or the flood risk assessment. In generally, I found this manuscript interesting, technically sound, and well organized. Nevertheless, I think this paper needs some revisions in order to clarify the novelty of the methods. I also suggest the authors to carefully review grammar and spelling throughout the entire manuscript.

The conclusion "SM is the main source of uncertainty for the stations with significant trend, while the DF contributed most to the uncertainty for the stations without clear

trend" only basin on two hydrological stations. I think this kind of conclusions should be based on statistical results. Therefore, I strongly recommend either presenting a strong argument in favor of only two stations or better using higher-number stations.

Also, the method of DPOT is not explained clearly, I do not understand why the authors choose only the station with significant downward trend, what if the series have a significant increasing trend? Wound the DPOT also be a better sample method to reduce the uncertainty?

The authors should clarify why the Chi-squared method was selected for fit test.

The abbreviations of distribution is inconsistent in the manuscript. Please check and correct.

In the abstract: "But uncertainties. . .." should replace by "though uncertainties".

There lots of mistakes in the superscripts of units in section 2

I suggest the authors provide further practical interpretation of the results presented in the last section.

All the references should be edited according to the format of HESS. The reference listed below was not cited in the manuscript. Kianfar, B., Fatichi, S., Paschalis, A., Maurer, M., and Molnar, P.: Climate change and uncertainty in high-resolution rainfall extremes, Hydrol. Earth Syst. Sci. Discuss., doi:10.5194/hess-2016-536, 2016

---

## Author Comment (AC1) · 18 May 2017

Thank you for the review and constructive comments. We have responded item by item as below to your comments.

This paper introduces and describes a quantitative way to assess the uncertainty in estimating return periods by considering different data sampling methods, distribution functions and parameterizations estimation methods. The topic of this manuscript had practical value for the engineering design or the flood risk assessment. In generally, I found this manuscript interesting, technically sound, and well organized. Nevertheless, I think this paper needs some revisions in order to clarify the novelty of the methods. I also suggest the authors to carefully review grammar and spelling throughout the entire

[Figure]

manuscript. I have provided some editorial suggestions at the end of this review, but I may have missed some.

Q1, The conclusion "SM is the main source of uncertainty for the stations with significant trend, while the DF contributed most to the uncertainty for the stations without clear trend" only basin on two hydrological stations. I think this kind of conclusions should be based on statistical results. Therefore, I strongly recommend either presenting a strong argument in favor of only two stations or better using higher-number stations. Answer 1: We prefer to add more theoretical analysis in conclusion and discussion parts. Section 3.1 shows that the data series is very sensitive to the return level estimation, especially for the series which have significant trend. Both of the variations of skewness and kurtosis between three sample series are larger for the series with significant trend than those without. That is to say, return level of three sample series have lager range for the series with significant trend than those without. In revised manuscript, we conducted a sensitivity test to figure out whether the results from this study transferable to other stations under different climate conditions. By using a detrend method, we generated a new discharge series, and found that the main uncertainty to the estimation of return levels is from distribution functions.

Q2, Also, the method of DPOT is not explained clearly, I do not understand why the authors choose only the station with significant downward trend, what if the series have a significant increasing trend? Wound the DPOT also be a better sample method to reduce the uncertainty? Answer 2: We know that there is periodicity for a discharge series usually. If the observed record was in a peak or positive period of the periodicity, return levels might be heavily overestimated. The alternative is the DPOT sampling method introduced in this study. Of course, we will describe the DPOT method more clearly and understandable.

Q3, The authors should clarify why the Chi-squared method was selected for fit test. Answer 3: Chi-squared is a most commonly used goodness of fit test. The advantage and applicability of this method will be added in the method section.

Q4, The abbreviations of distribution are inconsistent in the manuscript. Please check and correct. Answer 4: Sorry for our carelessness. We corrected all abbreviations to make sure acronyms are consistent throughout the paper.

Q5, In the abstract: "But uncertainties..." should replace by "though uncertainties". Answer 5: We have corrected this sentence.

Q6, There lots of mistakes in the superscripts of units in section 2 Answer 6: We have checked and corrected all of them.

Q7, I suggest the authors provide further practical interpretation of the results presented in the last section. Answer 7: We will enrich the practical interpretation of our results in the discussion parts. In fact, we analyzed the uncertainty sources of return level estimation, and delivered the possible way to reduce the uncertainties in this paper. Our results could be useful for appropriate assessment of disaster and the construction of hydraulic projects such as dams, bridges, and pipelines (Rosbjerg and Madsen, 1998; Milly et al., 2002; Cooley et al., 2007; Salvadori et al., 2011; Rootzén and Katz, 2013).

Q8, All the references should be edited according to the format of HESS. The reference listed below was not cited in the manuscript. Kianfar, B., Fatichi, S., Paschalis, A., Maurer, M., and Molnar, P.: Climate change and uncertainty in high-resolution rainfall extremes, Hydrol. Earth Syst. Sci. Discuss., doi:10.5194/hess-2016-536, 2016 Answer 8: We corrected the reference format.

---

## Author Comment (AC2) · 18 May 2017

Thank you for the review and constructive comments. We have responded item by item as below to your comments in italic.

This paper addresses the uncertainty in estimating return periods by considering different data sampling approaches, distribution assumptions and parameter estimation methods. It showed that different approaches could lead to very different results, and the optimal approach varies across gauging stations. The work is technically sound and the manuscript is clearly organized. The results are of practical importance.

Major comments:

Q1, The study somewhat lacks an in-depth discussion. The results are case-dependent and do not have a general implication. One reason is that it considered only two gauging stations, and they are in the similar climate and watershed conditions. They do differ in the flow variation trend, the uncertainty results reflect the difference. However, the author didn't go further to reveal de underlying physical or mathematical reasons for the difference. Thus, no general conclusion can be drawn from the comparison. I improve the scientific significance of this work, I suggest the authors either analyze more stations in different watershed and climate conditions, or provide a theoretical analysis of the difference between the two stations.

Answer 1: We focus mainly on the contributions of the different sources to the uncertainties of estimated return levels for the discharge series with or without significant trends. Section 3.1 shows that the data series is very sensitive to the return level estimation, especially for the series which have significant trend. Both of the variations of skewness and kurtosis between three sample series are larger for the series with significant trend than those without. That is to say, return level of three sample series have lager range for the series with significant trend than those without.

We prefer to add more theoretical analysis in conclusion and discussion parts. In revised manuscript, we conducted a sensitivity test to figure out whether the results from this study transferable to other stations under different climate conditions. By using a detrend method, we generated a new discharge series, and found that the main uncertainty to the estimation of return levels is from distribution functions.

Q2, The introduction to the sampling methods in Section 2.2 is two succinct. With the limited information, readers may not be able to understand how the extreme series are actually produced through POT and DPOT. Missing such critical information makes it hard for readers to understand the work.

Answer 2: We will add formula and references to revise the manuscript to describe all processes including the POT and the DPOT sampling methods more clear.

[Figure]

Minor comments:

Q3, Acronyms are not consistent in the text and figures. Some examples are: MLE vs. ML; P III vs. P3; LN vs. LN2; Gam vs. GAM; GUM vs. GUM. . .

Answer 3: Sorry for our carelessness. We corrected all abbreviations to make sure acronyms are consistent throughout the paper.

Q4, Section 3.3 was poorly written. It pours a lot of numbers here, but provides few insights. This section could be condensed into a couple of tables or figures, following by a paragraph or summary.

Answer 4: This part will be rewritten to highlight the main topic.

Q5, Please check the units, many of them do not have correct superscripts.

Answer 5: We will check and correct all of them.

Q6, The introduction should articulate the research objectives.

Answer 6: Main aim of this paper is to quantity the uncertainty sources of return level estimation by adapting the ANOVA method to two hydrological stations which have long-term observational records. Unlike previous publications on this topic, we focused on the return level variation caused not only by the choice of distribution functions, but also by the other two uncertainty sources: sampling method and parameterization method. As a result, possible way to reduce the uncertainties of return level estimation is delivered in this study. We believe our findings are meaningful for decision makers to get optimistic results. We will articulate the research objectives in the end of introduction section.